# Cryptococcus in Wildlife and Free-Living Mammals

**DOI:** 10.3390/jof7010029

**Published:** 2021-01-06

**Authors:** Patrizia Danesi, Christian Falcaro, Laura J. Schmertmann, Luisa Helena Monteiro de Miranda, Mark Krockenberger, Richard Malik

**Affiliations:** 1Laboratory of Parasitology, Istituto Zooprofilattico Sperimentale delle Venezie, 35020 Legnaro, Padua, Italy; cfalcaro@izsvenezie.it; 2Veterinary Pathology Diagnostic Services, Sydney School of Veterinary Science, The University of Sydney, Sydney 2006, Australia; laura.schmertmann@gmail.com (L.J.S.); luisahmiranda@gmail.com (L.H.M.d.M.); mark.krockenberger@sydney.edu.au (M.K.); 3Centre for Veterinary Education, The University of Sydney, Sydney 2006, Australia; richard.malik@sydney.edu.au

**Keywords:** *Cryptococcus*, cryptococcosis, felid, marine, koala, wildlife, *C. neoformans*, *C. gattii*

## Abstract

Cryptococcosis is typically a sporadic disease that affects a broad range of animal species globally. Disease is a consequence of infection with members of the *Cryptococcus neoformans* or *Cryptococcus gattii* species complexes. Although cryptococcosis in many domestic animals has been relatively well-characterized, free-living wildlife animal species are often neglected in the literature outside of occasional case reports. This review summarizes the clinical presentation, pathological findings and potential underlying causes of cryptococcosis in various other animals, including terrestrial wildlife species and marine mammals. The evaluation of the available literature supports the hypothesis that anatomy (particularly of the respiratory tract), behavior and environmental exposures of animals play vital roles in the outcome of host–pathogen–environment interactions resulting in different clinical scenarios. Key examples range from koalas, which exhibit primarily *C. gattii* species complex disease presumably due to their behavior and environmental exposure to eucalypts, to cetaceans, which show predominantly pulmonary lesions due to their unique respiratory anatomy. Understanding the factors at play in each clinical scenario is a powerful investigative tool, as wildlife species may act as disease sentinels.

## 1. Introduction

Cryptococcosis in wildlife and domesticated ungulates has been well reported but has not been systematically reviewed. This paper aims to correct this deficiency in the literature, focusing on free-living wild animals and those kept in captivity. A separate publication will focus on cryptococcosis in production animals (sheep, goats, camelids) and horses. Cryptococcosis is usually a sporadic disease in individual animals. Occasionally several animals are affected at the same time in the same place. Overall, it is an uncommon problem throughout the world. As an entity, it is more common under certain geographical conditions, e.g., in certain parts of Australia, Brazil and the Pacific North West of Canada and the United States of America (USA). The range of animals susceptible to infection is extraordinarily wide—from single-celled *Acanthamoeba* to large marine mammals. This includes a wide diversity of wildlife species, small companion animals, horses, production animals and human patients. In addition to koalas and companion animals (cats, dogs, ferrets), cryptococcosis has been described both as a cause of clinical disease, subclinical infection and/or asymptomatic mucosal colonization in many terrestrial and aquatic placental mammals, including marsupials, monotremes, birds, reptiles, amphibia and fish.

The overall pattern of cryptococcosis in wildlife species is complicated. Clinical and investigative findings from populations of wild animals are disparate and scattered over many different journals, with few systematic studies of large numbers of cases, so the challenge is to develop a conceptual framework that allows these observations to contribute to our understanding of this disease in all species. The fragmentary nature of the existing literature ranges from single case reports due to opportunistic necropsy examinations after accidental death, to more detailed case studies of patients living in artificial captive environments (e.g., koalas in captivity in Australia, USA, Spain and Japan) or involved in outbreak events (e.g., in Vancouver Island and nearby British Columbia; epizootics affecting goats in Spain). Complicating this fragmented literature, the accuracy of pathogen identification is influenced by which diagnostic specimens were collected and by the variable application of available laboratory tools (i.e., cytology, histopathology, culture, MALDI-TOF MS, multilocus sequence typing (MLST) and whole-genome sequencing), but also by taxonomic reassessment of the pathogens over time. The *Cryptococcus neoformans* and *C. gattii* species complexes are the most important causes of cryptococcosis in wildlife, with higher prevalence in animals from endemic areas (Western Canada, Australia and Brazil), and especially in koalas, an arboreal marsupial which live in eucalypt trees strongly associated with *C. gattii*.

Additional epidemiologic information is afforded by the systematic monitoring of many wild and synanthropic animal populations, usually performed at the same time as other public health actions (e.g., avian influenza surveillance in birds and culling of an overabundant pest species) or animal management programs (e.g., annual health checks of vulnerable animal populations). In general, results from studies of this type suggest that wild animals are more likely to have subclinical disease (i.e., infection without overt signs) than to be clinically affected by cryptococcosis [1,2].

This review aims to summarize information about cryptococcosis in wild animals including feral ungulates, terrestrial species and marine mammals which have in common a ”free-living lifestyle”. Therefore, we have included reports concerning animals living in undomesticated settings (i.e., wild animals) in their natural environment, animals living in a confined area such as zoological parks (e.g., koalas, marine mammals, large felids) and, briefly, ungulates living in an open range environment.

## 2. Nomenclature

The nomenclature of *C. neoformans* and *C. gattii* species complexes remains controversial and has the potential to be confusing, as there have been numerous taxonomic changes over the past 40 years and the taxonomy continues to be in a state of flux, with two distinct taxonomic ”camps” [3].

Historically, pathogenic *Cryptococcus* isolates were treated as a single species (*C. neoformans*) for more than a century [4]. Heterogeneity among isolates became increasingly apparent from the 1960s and led to the recognition of four serotypes (A, B, C, D) based on capsular polysaccharide epitopes [5]. The discovery of two different teleomorphs, *Filobasidiella neoformans* (anamorph; *Cryptococcus neoformans*) and *Filobasidiella bacillospora* (anamorph; *Cryptococcus gattii*) [6,7], confirmed a division of the previously recognized single species, and this taxonomic revision was later verified by whole-genome sequencing studies [8]. Thus, in 2002, isolates belonging to serotypes B and C were formally classified as *C. gattii* [9], while *C. neoformans* encompassed all serotype A, AD and D strains [10].

Over the last 20 years or so, population structure analysis of the two species using molecular typing methods, including PCR fingerprinting [11], amplified fragment length polymorphism (AFLP) analysis [12], multilocus sequencing MLST [13] and whole-genome sequencing [14,15], has demonstrated that *C. neoformans* and *C. gattii* isolates both comprise multiple genetically diverse monophyletic clades [16,17]. A recent proposal was made to designate seven MLST clades identified among *C. neoformans* and *C. gattii* into new species: *C. neoformans* into two species and *C. gattii* into five species (Figure 1) [18]. To date, the newly proposed nomenclature has not achieved acceptance within the scientific community (debated at the 10th International Conference on *Cryptococcus* and Cryptococcosis in Brazil, 2017), due to unclear biological differences amongst the lineages and a lack of clear consensus concerning the limits and numbers of putative species boundaries. In order to avoid confusion, in this review, we refer to ”*Cryptococcus neoformans* species complex” and ”*Cryptococcus gattii* species complex” to identify these fungal pathogens, terms that are well understood and accepted amongst mycologists, veterinarians and physicians, while consensus of further species delineation is achieved, although there remains uneven adoption of the proposed taxonomic nomenclature resulting in unwarranted confusion in the literature [3].

To make reading this manuscript simpler, we will use anamorph names and molecular types as reported in Figure 1. 

## 3. Epidemiology

Members of the *C. neoformans* species complex and members of the *C. gattii* species complex are both globally distributed, although there are important differences in their environmental associations and with their specific geographical preferences [19].

*C. neoformans* tends to be widely distributed throughout the environment, and although tree and plant associations are documented throughout the world, traditionally, the strongest environmental magnifier is aged dried avian guano accumulations protected from light (particularly pigeon excreta) [20].

*C. gattii* has tended to have a much more restricted geographic distribution than *C. neoformans*, causing human disease in temperate to tropical climates in Australia, Papua New Guinea, South-East Asia, the Indian subcontinent, parts of Africa, Mexico, Brazil, Paraguay and California. The global distribution of *C. gattii* has been related to tectonic plate movements [21,22].

The first environmental isolation of *C. gattii* was made by David Ellis and Tania Pfeiffer, in the Barossa Valley of South Australia. These investigators established the remarkably specific ecological association with *Eucalyptus camaldulensis* (river redgum), widely distributed along rivers in mainland Eastern Australia. The global distribution of *C. gattii* was long thought to reflect the global distribution of eucalypt natural habitat and forestry operations; however, recently, carob and olive trees were suggested as additional important environmental niches for *C. gattii* strains in the Mediterranean basin [23].

Unlike *C. neoformans*, other *Cryptococcus* species, including *C. uniguttulatus*, *C. albidus* and *Papiliotrema laurentii* (formerly *Cryptococcus laurentii*), are commonly isolated from excreta and cloacal swabs from feral pigeons (*Columba livia*) and birds of Charadriiformes, Anseriformes and Gruiformes orders [2,24,25]. Worldwide, plant and animal associations play a key role in the epidemiology of cryptococcosis [26,27]. A substantial difference exists between the *C. neoformans* and *C. gattii* species complex [21]. Although isolates of *C. neoformans* species complex occur more commonly in immunosuppressed human patients than isolates of the *C. gattii* species complex [28], each can cause disease in apparently immune competent humans and other animals [29].

Within the *C. neoformans* species complex, *C. neoformans* VNI is most common amongst both human and companion animal patients (dogs, cats, ferrets). In contrast, VNII is rarer, associated with higher morbidity and has an association with HIV-positive human patients. In companion animals, the consensus view is that any association of cryptococcosis caused by *C. neoformans* with reduced immune status is weak [29], although there is some suggestion of an impact of retroviral status and genomic predispositions on disease outcomes in cats, with an increased likelihood of severe cryptococcosis in cats infected by feline leukemia virus and/or feline immunodeficiency virus [30] and higher prevalence and more severe and refractory disease in Ragdoll and Birman breeds.

## 4. Pathogenesis and Virulence Factors

Important virulence factors of the *C. neoformans* and *C. gattii* species complexes include (i) the formation of a thick polysaccharide capsule to avoid and suppress the host immune response, (ii) capacity to grow at mammalian body temperature, (iii) production of melanin by laccase to resist host and environmental free radicals, (iv) production of urease and phospholipase B to facilitate tissue invasion, (v) production of superoxide dismutase and trehalose to evade the host immune response and (vi) development of polyploid ”titan cells” to further resist phagocytosis [31]. These virulence factors are also important features for environmental survival, including survival within soil amoeba, and evolution has favored the development of features which permit survival in the environment, which also facilitate survival in mammalian hosts [32]. 

The evolution of virulence in *Cryptococcus* and various other mycotic pathogens may have been driven by interactions and predation by free living amoeba species such as *Acanthamoeba castellanii*. This proposed evolutionary process has been put forward as the ”amoeboid predator-fungal animal virulence” hypothesis [33]. Since the 1950s, it has been known that *A. castellanii* feeds voraciously on *C. neoformans* [34]. The cryptococcal polysaccharide capsule, melanin synthesis and phospholipase have each been shown to be essential for *C. neoformans* to resist predation by *A. castellanii*, and *C. neoformans* responds to both *A. castellanii* and mammalian macrophages by enlarging, triggered by expression of phospholipase-B [35]. 

Because the *Cryptococcus* species complexes are present ubiquitously in the environment, exposure typically occurs early in life, presumably through inhalation of spores or desiccated yeast cells. Such exposure is typically asymptomatic, as the fungus is either cleared, colonizes the respiratory or alimentary tract transiently or becomes dormant [36]. In immunosuppressed individuals, however, fungi can reactivate and disseminate hematogenously from the lungs and the hilar lymph nodes. Although this pathogen can affect virtually any tissue, it has a striking tropism for the central nervous system (CNS), causing meningoencephalitis that is fatal, without treatment [37], with similar predilection for neural ocular tissues. 

Many critical fungal pathogens of plants, animals and humans can switch between a unicellular yeast morphology and a multicellular hyphal form. Such morphological transitions are associated with an ability to infect, invade, evade and disseminate widely. Like other dimorphic pathogens acquired from the environment, such as *Histoplasma*, *Blastomyces* and *Coccidioides*, *Cryptococcus* species complexes are found as a yeast form in host tissues [38] (Figure 2). Variations in cryptococcal life cycle and the associated cellular adaptions profoundly influence cryptococcal pathogenesis [39]. Firstly, different morphotypes elicit distinct host responses [40]. Secondly, sexual and asexual cycles generate infectious basidiospores that can penetrate deeply into the bronchial tree [41,42]. Finally, the phenotypic and genotypic diversity referable to sexual reproduction increases cryptococcal fitness and adaptability [42,43,44]. 

## 5. A Possible Environmental Niche

Although *C. neoformans* species complex fungi are often associated with pigeon guano, birds are most unlikely to be the major source of cryptococci in the natural world, since only low *C. neoformans* prevalence, ranging from 0 to 2.2%, was found in samples from crop, feet and cloacal swab of Columbiformes and water birds in Italy, Sweden, Czech republic and Iran [2,25,45,46]. Most likely, *Cryptococcus* propagules from the environment are deposited onto the nitrogen-rich guano where they grow, rather than being transferred to it through a passage along the alimentary tract of the bird. Indeed, the core body temperature of birds is as high as 42–44 °C, which is inhibitory to cryptococcal growth and multiplication; the high ammonia concentrations in fresh droppings are also inhibitory to growth. It is of great interest that *C. neoformans* can complete its life cycle, including mating, on pigeon guano although *C. gattii* complex fungi do not. Thus, it seems quite possible that pigeon guano could represent the definitive ecological niche of *C. neoformans* [47]. In contrast, no specific strong association with bird, animal or insect vectors has been established.

The *C. gattii* species complex (mainly VGI) was well known as an important pathogen of Australian wildlife, although its close relative *C. gattii* VGII also proved to be an important cause of disease of wildlife on Vancouver Island and the Pacific Northwest, while most recently, an association between *C. gattii* VGV and the African hyrax has been shown in Zambia. *C. gattii* VGI is the most geographically widespread member of this species complex in Australia [28], and it is by far the most common cause of cryptococcosis in Australian wildlife [48]. This is most likely because of the strong association of VGI with *Eucalyptus* spp. trees [43]. 

A *C. gattii* VGII outbreak that began in 1999 on Vancouver Island, and subsequently became endemic, caused numerous human and animal infections. Furthermore, the pathogen spread to the nearby mainland of British Columbia. Since 2006, the molecular type VGIIc emerged in human and veterinary cases in Washington and Oregon in the Pacific Northwest of the USA [49]. Over the 1999–2003 period, there was a progressive increase in the annual occurrence of animal cases diagnosed, while human cases plateaued in later years. Critically, there were approximately 75% more animal cases than human patients, and often in atypical animal species, such as horses, ferrets and marine mammals (rather than just cats and the occasional dog). This is despite the fact that in all likelihood, animal cases were more likely to go undiagnosed or unreported compared to humans [50]. Animal cryptococcosis cases had been identified on Vancouver Island prior to 1999 but at a much lower prevalence. Therefore, the suggestion is that the organism emerged in the region prior to its identification as a causative agent for human disease. These findings imply that animals, by virtue of (i) increased case numbers, (ii) emergence in unusual species and, potentially, (iii) a shorter incubation period, may serve as outstanding sentinels for human cryptococcosis. Their role in establishing the emergence of *C. gattii* in Canada should receive greater emphasis [51,52] as should the contribution of the local veterinary pathologist, Dr Sally Lester, whose diagnostic observations resulted in a timely appreciation of the outbreak. 

In 2019, the first isolates representing the VGV lineage were recovered, together with VGI and VGII isolates from middens, midden soil and tree holes of the southern tree hyrax (*Dendrohyrax arboreus*) [53]. While *C. gattii* has not yet been demonstrated to pass through the alimentary tract of mammals, associations with small mammals such as hyrax suggest a potential evolutionary mechanism to generate adaptations that confer pathogenicity—a notion recently posited as the “endozoan, small-mammal reservoir hypothesis” [54]. This concept deserves to be further explored.

The southern tree hyrax is a small herbivore, counterintuitively closely related to the elephant. Hyraxes defecate in communal latrines situated in the crevices of rocky kopjes. These sites are used over numerous generations [55], being located in sheltered rocky caves where droppings are likely to accumulate for periods of upwards of 50,000 years forming middens of stable paleoenvironmental urea-rich nitrogenous material containing an unprecedented richness of paleoclimatic proxies [56]. *Cryptococcus* species complexes fungi have a marked tropism for urea as a nutrient substrate, including pigeon guano, which is known to support the prolific growth of *C. neoformans* and to a lesser extent *C. gattii* [47]. The finding that hyrax middens are hotspots of *Cryptococcus* diversity (including VGI, VGII and VGV from sampling in Zambia) suggests that their ecological stability in regions rich in nitrogen availability will result in them being vital crucibles for cryptococcal evolution. Studying these environments will likely provide a fertile ground for further discovery of diversity within this genus [27].

## 6. Wild Mammals

### 6.1. Cryptococcosis in Wild Felids

Among wild felids, cryptococcosis is described only in cheetahs. Most of the cases have been recorded from South Africa. Only a single case was described from Cuba in a cheetah following reactivation of infection probably acquired 16 months earlier, at the time the animal was moved from South Africa [57]. Clinical signs described in cheetahs included hind limb lameness [58], a skin mass overlying the zygomatic arch with enlargement of ipsilateral regional lymph node [59] and generalized weakness. Neurological signs described in cheetahs with cryptococcosis include incoordination and change in behavior, and ataxia in all four limbs [59,60]. (Figure 3, Table 1).

Most animals showed concurrent pulmonary and CNS signs [58,59,60]. Inhalation of airborne organisms with subsequent hematogenous or lymphatic spread was the most likely route for cryptococcal infection. The pulmonary cryptococcoma present in these cheetahs provides strong circumstantial evidence for this theory. Interestingly, in this species, the upper respiratory tract is spared, suggesting the deep inhalation during and after exertion, with deep tidal volumes, might facilitate deposition of infective propagules deep into the bronchial tree. Cryptococci from the lungs spread to the CNS, either hematogenously or by local extension through the cribriform plate (in rare cases of sinonasal infection). In the absence of apparent upper respiratory disease in these cheetahs, it is likely that dissemination was typically hematogenous from primary lung lesions [58,59,60]. These animals were seronegative for FeLV, FIV, feline coronavirus and paramyxovirus.

Of great interest was the case of *C. gattii* VGI described in Cuba from a cheetah imported 16 months earlier from South Africa [57,61]. The authors posit that the animal had a dormant infection at the time of translocation to the zoo in Cuba. Considering the accepted notion that there is a regional correlation between environmental and clinical isolates [64], there is no evidence whatsoever for *C. gattii* isolates in Cuba from clinical and environmental samples, despite an extensive search [65]. 

Even though cryptococcosis is the most common systemic mycotic disease described in felids, observations from other species suggest that subclinical infection with spontaneous elimination of cryptococci is probably much more common than clinical infection. Factors favoring progression to disease have been linked with the presence of comorbidities or other immune-suppressive conditions. In domestic cats, there is anecdotal evidence for genetic susceptibility to cryptococcosis [66]. The cheetah can be considered to be conceptually akin to particular pedigree cat breeds in terms of vulnerability to certain infectious diseases because the entire species lacks heterogeneity at MHC loci encoding peptides mediating immune responsiveness to pathogens [58,67]. Other studies have reached a different conclusion, namely, that immune status in cheetahs depends on mechanisms not solely related to their lack of genetic diversity [67].

A common necropsy finding in captive cheetahs is diffuse adrenocortical hyperplasia. This change has been described in many other captive wild animals [68,69,70]. For example, the extent of adrenocortical hyperplasia has been correlated to the stress of captivity in nine-banded armadillos, the platypus and harbor porpoises [68,69,71]. High blood cortisol concentrations occur in animals with diffuse adrenocortical hyperplasia and this, together with the well-known depression of monocyte function by cortisol, may go some way to explain the predisposition of the captive cheetah to develop cryptococcosis when kept under stressful conditions.

In addition, lymphocytic depletion of the spleen has also been observed in some captive cheetah cases [70]. These immunosuppressive effects would be enhanced by the polysaccharide capsule of *C. neoformans*, which inhibits phagocytosis, plasma cell function and leukocyte migration [59,62,63]. Along with domestic felids, adult male cheetahs appear to have the highest incidence of cryptococcosis amongst the *Felidae* [58,59,60].

### 6.2. Cryptococcosis in Koalas

Cryptococcosis occurs sporadically in a broad range of Australian native mammals. There is a preponderance of respiratory and neurological presentations amongst these cases, with tracheobronchial pneumonia and sinonasal disease also being common. Disseminated disease and disease affecting other organs or tissues is less frequent, although occasionally infections seem to occur after penetrating injury and can involve subcutaneous tissues and underlying bone [72] (Figure 4). The prevalence in koalas is comparatively high, probably the highest prevalence of any species (3% of koala necropsies are attributable to cryptococcosis), in part because they are commonly held in wildlife parks and zoological gardens where they appear to amplify the presence of *C. gattii*, both VGI and VGII, in their immediate environment. Additionally, koalas are more commonly intensively studied in wild and captive populations, and they have an intimate association with a common environmental niche of *C. gattii* VGI, viz. a variety of eucalypt tree species. Consequently, the disease is well recognized and thoroughly characterized in this species [73,74,75].

In koalas, as in most other species, respiratory tract involvement is observed most commonly (78% of cases), with pneumonia (73% of respiratory tract cases) and upper respiratory tract disease both common (48% of respiratory tract cases). A substantial proportion of koalas with upper respiratory tract lesions (mostly sinonasal and nasopharyngeal disease) also have lower respiratory tract involvement (42% of cases with sinonasal disease also have pneumonia). Spread of infection to regional lymph nodes is commonly present (20% of cases), but rarely without additional tissue involvement. Disseminated disease is relatively frequent (34% of cases) and can develop after a longstanding respiratory illness (76% of disseminated cryptococcosis has an identifiable primary focus in the respiratory tract; Figure 2). As in other species, cryptococcosis can be regarded as relatively neurotropic (32% of cases have CNS involvement), but less commonly as the only organ system reportedly involved (14% of koala cryptococcosis cases).

Although cryptococcosis is commonly observed and documented in captive koalas, case clusters in free-ranging populations are being recognized more commonly (39% of cases documented have been reported in free-ranging koalas). There is little evidence to suggest a gender bias of koala cases, with 47% of cases in males and 53% in females. The role of potential immune suppression or association with koala retrovirus variants in the pathogenesis of cryptococcosis remains unclear but intriguing, and the subject of active investigations [76]. 

For cryptococcosis to develop, contact with the organism is necessary for colonization to occur. Once colonization is established, subclinical or clinical cryptococcosis may then develop. Progression may be rapid or slow, but is not inevitable, and colonization and even subclinical disease may spontaneously resolve over time as a result of effective adaptive immunity. Spontaneous resolution after transient subclinical disease is a common occurrence in young koalas soon after independence.

Most koala cases where the causative agent was definitively identified were attributable to the *C. gattii* species complex [77], with a single case caused by the *C. neoformans* species complex in a captive koala in Spain [75]. In most of Australia, especially along the east coast, cryptococcosis is usually caused by *C. gattii* VGI, although in south-western Western Australia (WA) infections can also involve *C. gattii* VGII, including cases where koalas have been infected in WA and developed disease following translocation elsewhere within Australia [78].

Colonization, subclinical and clinical cryptococcosis in captive koalas presents a substantial management concern in most koala facilities, especially in relation to translocation of animals, and serological monitoring (using the latex cryptococcal antigen agglutination test or lateral flow assays) is generally recommended as a component of a standard health check at least once or twice a year, and prior to moving koalas to new facilities. Mycological culture of the nasal mucosa may sometimes be added to monitoring programs to understand the degree of environmental contamination and exposure of the resident koalas to the organisms and a predictor of the prevalence of subclinical and clinical cryptococcosis [48].

### 6.3. Cryptococcosis in Marine Mammals

Marine mammals, which include delphinids (*Stenella*, *Tursiops* and *Lagenorhynchus* species), porpoises (*Phocoenoides dalli* and *P. phocoena*), pinnipeds such as harbor seal (*Phoca vitulina*) and sea lion (*Zalophus californianus*) and baleen whales such as the southern right whale (*Eubalaena australis*) are susceptible hosts for various mycoses including cryptococcosis (Table 2). Marine mammals are likely to be at risk for developing infection only when they come sufficiently close to land to be exposed to infective propagules (likely spores) in effluent and run off that is washed into the ocean. Animals with blow holes are especially vulnerable, as they breathe in an enormous tidal volume with marked inspiratory effort following explosive expiration, and without the benefit of filtration of inspired air via sinonasal protective mechanisms. As a result, a large inoculum of infective spores can be carried deep into the lower respiratory tract.

Since around 2000, cryptococcal infections have emerged as causes of disease in both humans and marine mammals living in the vicinity of Vancouver Island and the Pacific Northwest of the USA and Canada. Indeed, most of the 54 marine mammal cases summarized in Table 2 are from the Northwest Pacific region area, except for a spinner dolphin from Hawaii [79], a striped dolphin from Western Australia [80] and a whale neonate from South Africa [81]. 

During the emergence of *C. gattii* VGII in British Columbia, 22 juvenile harbor seals (*Phoca vitulina*) were monitored by using blow hole swabs (19×) and lung culture (3×), tested negative for *Cryptococcus* spp.

*C. gattii* was identified in 45/54 cases of symptomatic cryptococcosis (Table 2), *C. neoformans* in five animals, and the less pathogenic *Cryptococcus albidus* was reported from a stranded California sea lion (*Zalophus californianus*) [82]. The *Cryptococcus* species was not identified in three cases [80,83,84]. *C. gattii* VGI was genotyped from four delphinids [79,85] and a porpoise [86] and *C. gattii* VGIIa from three porpoises [85] and a single harbor seal [87].

A common finding in all marine mammal cases was the tropism for the lungs, lymph nodes and gastrointestinal tract, with the lung as the logical portal of entry for infection. Extension of the fungus to the brain was recorded in harbor seal (2 cases) and a harbor porpoise [87,90]. Interestingly, maternal–fetal transmission of *C. gattii* was described in a porpoise (*Phocoena phocoena*) [88]. In a single case, there was extension of the fungus into the stomach, adrenal glands, kidneys and spleen [79]. In the single case of whale neonate from South Africa, skin lesions were culture-positive for *Candida zeylanoides* but not for *C. neoformans*, which was identified only by PCR. Considering all marine mammal infections, porpoises (39) and dolphins (9) were the most common species infected. 

It should be stated, however, that there could be some bias in the observational data, if we just consider the reported fungal cases in Table 2 as a true reflection of fungal diseases in marine mammals. Importantly, investigators have commented there is little doubt we are observing a more substantial number of cases in animals maintained in artificial vs. natural water systems. 

### 6.4. Cryptococcosis in Free-Living Ungulates

Among wild ungulates, only two cryptococcosis cases have been reported. The most recent case was a yearling white-tailed deer (*Odocoileus virginianus*) living in Nova Scotia, Canada [91]. This deer was found in the village of Greenwood in July 2014, exhibiting behavioral and neurological abnormalities. It had lost its fear of humans, was ataxic with a high-stepping gait, circling, and exhibited torticollis and a fixed vacant stare. Additional clinical signs included ptyalism with frothing from the mouth and dyspnea, with a gurgling respiration. The animal was euthanized and at necropsy, gross examination revealed multifocal masses that had effaced the normal architecture of the tracheobronchial lymph nodes and lung parenchyma. Lesions were absent from the liver, kidney and alimentary tract. A *Cryptococcus* species was isolated from a tracheobronchial lymph node aspirate and genotyped as *C. gattii* VGIIb by MLST typing [13]. 

The second case was an adult free-ranging, male Roosevelt elk (*Cervus elaphus roosevelti*), approximately two years-of-age from USA [92]. The animal had a two-day history of circling to the left. In the same area, two or three other elk with similar signs had been noticed over the previous two years; however, none of these animals had been submitted for postmortem examination. Since the elk exhibited several neurological signs and had the potential for coming into contact with domestically raised Rocky Mountain elk, it was culled by the local wildlife authorities. At necropsy, gross lesions were found only in the brain. Based on the histopathological and microbiological findings, a definitive diagnosis of meningoencephalitis and pneumonia caused by *C. neoformans* species complex was made. 

The white-tailed deer is a new host species for *C. gattii* in North America. Because this species is non-migratory, exhibiting only minor seasonal movements [93], this infection was considered autochthonous, indicating endemicity of the *C. gattii* VGIIb-like variant in Nova Scotia. This yet again highlights the value of non-migratory animals as sentinels for emerging infectious diseases [51,94].

Similarly, autochthonous cryptococcosis was described in small ruminants reared semi-extensively [95] or extensively in the Mediterranean area [96,97]. In goats, cryptococcosis developed as pulmonary and neurological disease mostly. Neurological signs, including ataxia, nystagmus, blindness, opisthotonos and progressive paralysis, have been described starting from few days prior to euthanasia or death and necropsy [96,98]. In the Mediterranean basin, human and animal cryptococcosis is mostly caused by *C. neoformans* species complex strains, yet *C. gattii* VGI was the cause of infection in the five caprine cryptococcosis outbreaks occurring over the period 1990–94 in Spain and over the 1994–2001 in Sardinia Island (Italy) [96,97]. These cases in sheep, goats, camelids and horses will be discussed in much greater detail in a subsequent review (Danesi and colleagues, in preparation).

Phylogenetic analysis demonstrated that the Italian *C. gattii* VGI isolates clustered more closely with environmental isolates from Alicante in the southeast Mediterranean coast of Spain than with those from other parts of Italy [96]. The authors’ speculation is that the typical goat breed, the Murciana, raised in the Murcia region (Alicante) but extensively exported across the Mediterranean area, including Sardinia, probably resulted in asymptomatic carrier goats (colonized by VGI) being responsible for the introduction of this molecular type of *C. gattii* to Sardinia. Such translocation events have been well demonstrated in koalas exported within Australia, or from Australia to Japan, and in people who have visited Vancouver Island from Europe [77,78,99]. 

More recently, the presence of *C. neoformans* and *C. gattii* species complexes in the Mediterranean environment in association with various tree species was supported by botanical investigations [23,100]. Various authors have suggested that trees might represent a significant environmental niche and a stable reservoir for both species. In contrast, bird guano might represent a secondary transient niche, especially for the *C. neoformans* species complex [23]. Other than eucalypt trees that have been present in the Mediterranean basin since the end of the 18th century [101], *C. gattii* VGI is most likely to be isolated additionally from Carob (*Ceratonia siliqua*) and Olive trees in a European setting [23,100]. The evidence that Carob trees are widely cultivated in all southern Mediterranean areas, Portugal, Italy and Spain, representing the most important producers worldwide, drives the hypothesis that they represent a crucial environment niche for animal exposure in these countries.

## 7. Discussion

This review has presented a great deal of information about cryptococcosis in a range of different species—with emphasis on disease features in wildlife, both free-living and in captivity, as opposed to the companion animals which tend to get the most attention in the veterinary literature (viz cat, dog, ferret, horse and birds) [29,102,103]. 

What then are the key messages for a mycologist interested in a holistic understanding of fungal disease pathogenesis?

For the authors, the most remarkable insights come from the appreciation of the importance of the interplay between the (i) mammalian host, (ii) the environment in which it lives and (iii) the nature of the cryptococcal species with the potential to colonize, invade and potentially cause subclinical or clinical disease.

In terms of the host, it comes down to size considerations, comparative anatomy (especially of the respiratory tract), body temperature and behavior. Large and relatively athletic animals such as goats, cheetahs and horses are more likely to have tracheal and lung involvement, as deep breaths during exercise might bypass filtering of the sinonasal cavity, permitting basidiospores to penetrate deep into the bronchial tree. In cats and ferrets, by way of contrast, filtration and turbulence make for better capture of spores by the ciliated respiratory mucosa of the nasal turbinates. However, most vulnerable of all species are the Cetacea that have lost the normal mammalian nasal filtering mechanism and instead breathe through a blowhole, taking large tidal volumes into the respiratory bronchioles with the deep inspirations which immediately precede a dive. It is for this reason that porpoises and dolphins are so susceptible to the acquisition of cryptococcosis in places with a high environmental presence of infective propagules, such as off the coast of Vancouver Island, resulting from heavy run-off of spores into the ocean. 

Body temperature is also a critical consideration. Cryptococcus species complexes fungi, as a generalization, prefer cooler temperatures, although they can survive at core mammalian body temperature. Thus, avian species that maintain core body temperatures of approximately 40 °C often have cryptococcal disease restricted to the upper respiratory tract, which is cooler, while birds such as the kiwi are much more vulnerable to disseminated disease as they have a much lower core body temperature. Behavior also has large impacts, which explains why arboreal koalas have the highest disease presence of all Australian mammals.

Clearly, an animal’s environment determines the extent to which there is exposure to the environmental niche of the fungus, and hence the range of inoculums to which an individual might be exposed. Therefore, an arboreal mammal such as a koala is more likely to be exposed to *C. gattii* VGI whose best-established niche is in detritus within well-developed tree hollows. As a result, cryptococcosis is more common in koalas than wombats, a closely related species which forages on the ground and lives in burrows. Instead, the wombat is exposed to alternate fungal pathogens such as *Emmonsia* [104]. What about the fascinating case clusters that have occurred affecting multiple animals simultaneously, such as flocks of sheep in WA [64,105] and the goats [96] in several parts of Spain? Presumably, there are environmental factors, such as the large stands of Australian eucalyptus trees in Spain, which favor an unusually heavy inoculum of infective propagules at certain times of the year. The environment in Vancouver Island likewise might be related to special features of that bioclimatic zone that have developed as a result of climate change and global warming.

Finally, cryptococcal biotype can be a critical consideration. *C. gattii* VGII, where it occurs, seems to be especially virulent. This manifests as a higher prevalence of disease in all species (including man and companion animals), and the appearance of overt cryptococcosis in unusual hosts, such as ferrets, horses and marine mammals. This is a feature of VGII equally in Vancouver Island, and in the southwest of Western Australia. 

The carriage of cryptococcal fungal elements is believed to be possible through long distances with posterior seeding in new geographic areas [106]. The travel of this pathogen via translocation of wildlife has also been described [107]. The fact that the disease can emerge long after the actual seeding and that a potential increase in the virulence is proposed is a strong argument for monitoring spontaneous cryptococcosis in the wildlife [106,108]. The surveillance of animal cases may flag areas of a higher risk for environmental transmission to humans and other animals as well as the emergence of new areas [109]. 

Therefore, the breadth of this review makes us focus on which factors are at play in a given clinical scenario—whether it is a flock of sheep with cryptococcosis in Western Australia, a cluster of dolphins off Vancouver Island or a collection of koalas in a wildlife park in Sydney or Perth. This is the reason why studying the occurrence of cryptococcal disease holistically by considering disease in all susceptible hosts is such a powerful investigative tool, as is common for some animal species to act as a disease sentinel in a given geographical location—due to either host factors, environmental factors or the biotope associated with that region. This interplay is illustrated graphically in Figure 5 [110].

## Figures and Tables

**Figure 1 jof-07-00029-f001:**
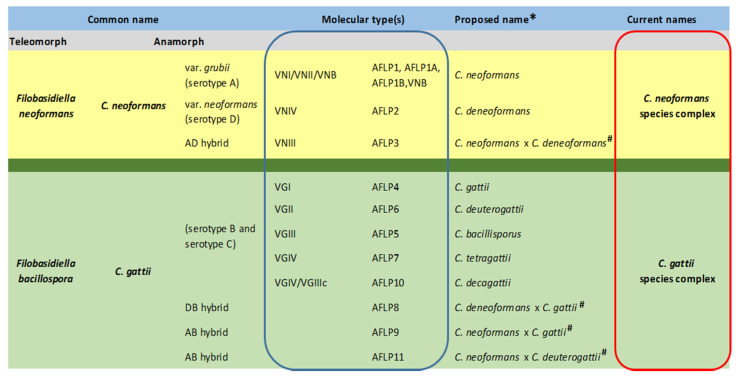
Historical nomenclature of *C. neoformans* and *C. gattii* species complex. * Hagen et al., 2015 [18]; ^#^ hybrid (Modified from Kwon-Chung et al., 2017 [3]).

**Figure 2 jof-07-00029-f002:**
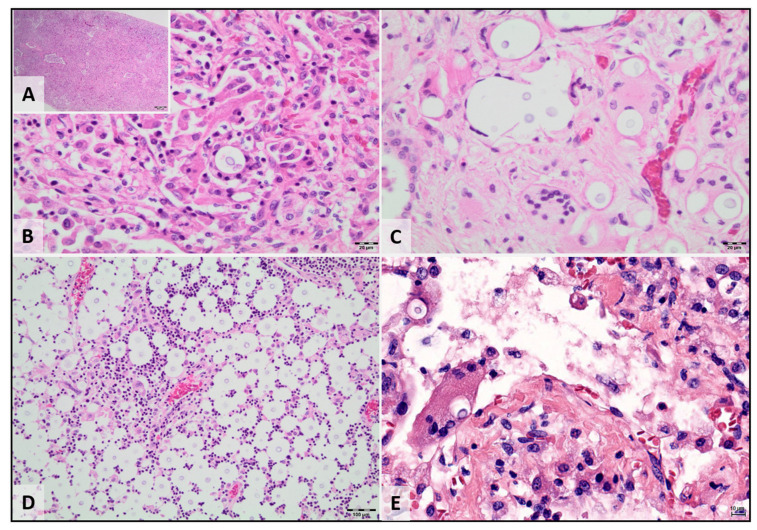
Cryptococcosis caused by *C. gattii* VGI in a 12 month old female koala (**A**–**D**) presenting with severe dyspnea and tachypnea and later developing seizures. (**A**) Lung at low magnification. There is marked consolidation of the pulmonary parenchyma. Some large airways containing exudate are evident. (**B**) The lungs exhibit a diffuse marked increase in cellularity due to an inflammatory infiltrate predominantly made up of macrophages. Note the budding yeast with the large negatively staining capsule. (**C**) In some parts of the specimen, the negatively stained capsule around the yeast is very large and there is an associated multinucleate giant cell macrophage response. (**D**) The thoracic lymph node exhibits a complete loss of normal architecture and is replaced almost entirely by large aggregates of encapsulated yeast cells, with some surrounding pyogranulomatous inflammatory infiltrate in places. (**E**) Lung at high magnification from a young male koala that presented with dyspnea, lymphadenomegaly and multifocal skin lesions associated with cryptococcosis caused by *C. gattii* VGI. There was significant widespread severe lung consolidation present at necropsy, in addition to multifocal lymph node involvement and multifocal skin lesions. Note the encapsulated yeast in the multinucleate giant cell macrophage.

**Figure 3 jof-07-00029-f003:**
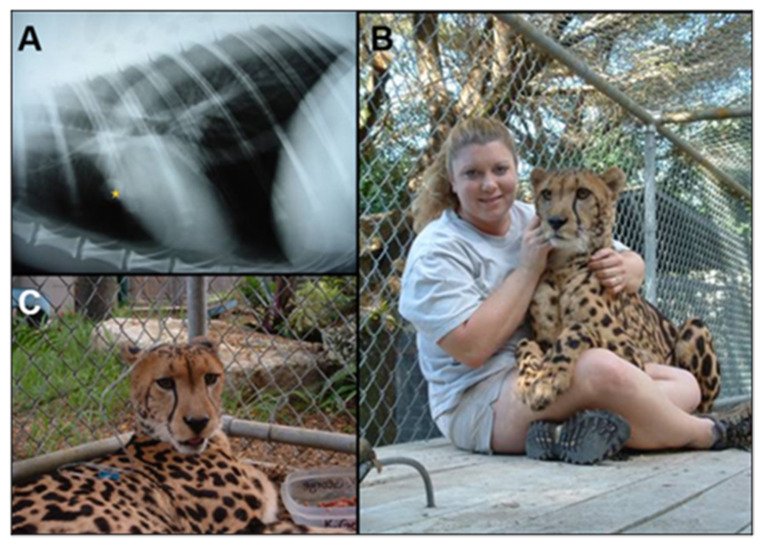
(**A**) Primary pulmonary cryptococcoma (yellow star) in a lateral chest radiograph from a captive cheetah. The “glassy-eyed stare” seen with peripheral blindness is evident in (**C**), which shows the cheetah being given a subcutaneous infusion of amphotericin B. The patient after successful therapy is shown in (**B**), with its carer; note the alert demeanor and normal posture.

**Figure 4 jof-07-00029-f004:**
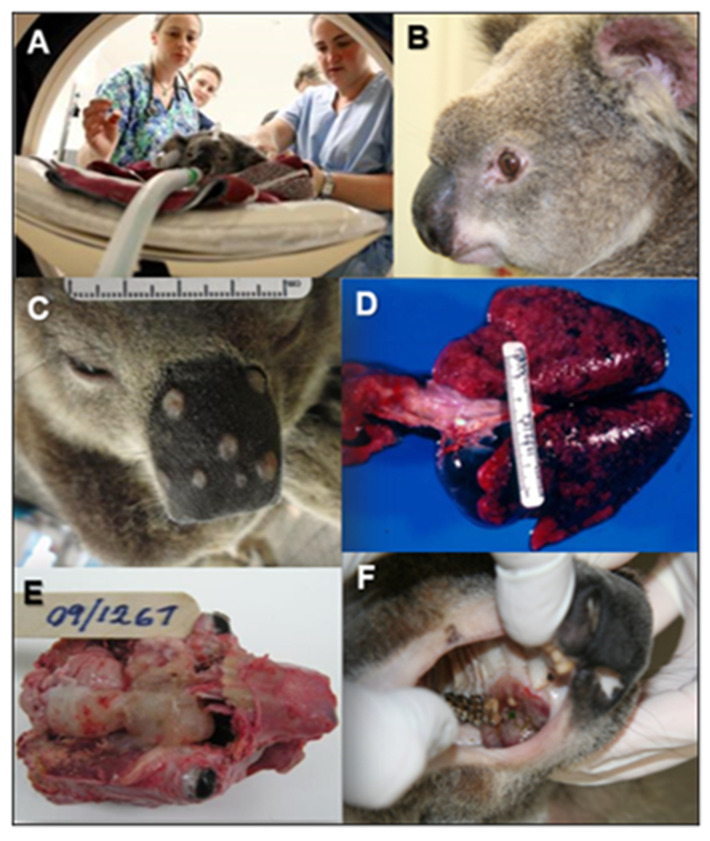
Cryptococcosis in the koala. (**A**) Koala with sinonasal cryptococcosis undergoing a computerized tomography scan. (**B**) Koala with destructive cryptococcal rhinitis due to *C. gattii*. (**C**) Multifocal skin involvement over the planum nasale of a koala with *C. gattii* infection. (**D**) Lungs from a koala that died of pneumonia due to *C. gattii* and *Bordetella bronchiseptica*. (**E**) Koala with cryptococcal rhinosinusitis with extension to the brain. (**F**) Koala with cryptococcal rhinitis with ventral extension to involve the hard palate.

**Figure 5 jof-07-00029-f005:**
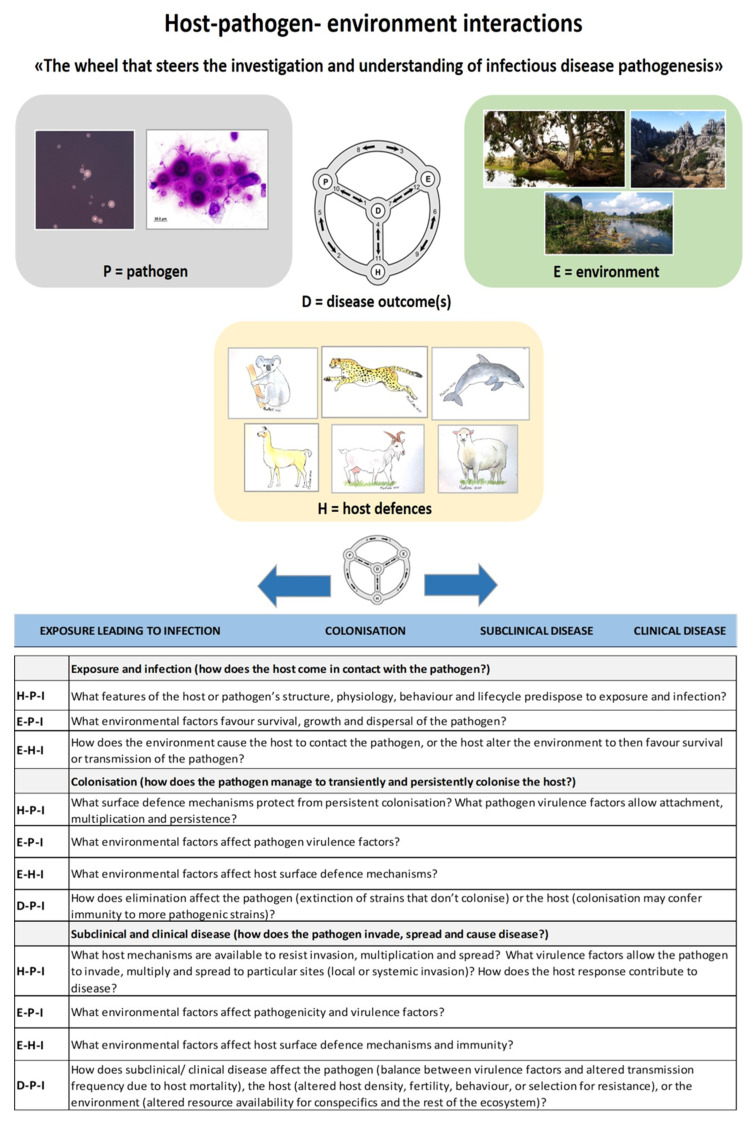
Host–pathogen–environment interactions: the wheel that steers investigation and understanding of infectious disease development. Interactions may enhance or ameliorate disease. **P**, pathogen; **H**, host defenses; **E**, environment; **D**, disease outcome (subclinical or clinical effect on the host). For any potential pathogen, the interaction of host, pathogen and the environment determine where on the disease–outcome continuum the relationship will lie. This integration of concepts gives rise to a template of questions and generates hypotheses for use in investigating wildlife disease (adapted from ideas developed by Emeritus Professor Paul Canfield).

**Table 1 jof-07-00029-t001:** Cryptococcosis in wild felids.

Case	Diagnosis	Species/Variety	Outcome (Treatment, Necropsy)	Location	Date	References
1	Pulmonary cryptococcoma and meningomyelitis (spinal cord)	*Cryptococcus gattii*	Necropsy	South Africa	2005	[58]
2	Pulmonary cryptococcoma and cryptococcal meningoencephalomyelitis	*Cryptococcus neoformans*	Necropsy	South Africa	1997	[60]
3	Pulmonary cryptococcomas and extensive meningoencephalomyelitis	*Cryptococcus neoformans*	Necropsy	South Africa	1999	[59]
4	Pulmonary cryptococcomas and extensive meningoencephalomyelitis	*Cryptococcus gattii*	Necropsy	South Africa	1999	[59]
5	Shortness of breath and nasal discharge	*Cryptococcus gattii*	The animal was taken to the veterinary clinic for sampling and died before any diagnosis could be made	Cuba	2010	[61]
6	Shortness of breath and nasal discharge	*Cryptococcus gattii* AFLP4/VGI		Cuba	2011	[57]
7	Nasal cryptococcosis	Abstract not available		USA	1982	[62]
8	Osteomyelitis	Abstract not available				[63]

**Table 2 jof-07-00029-t002:** Cryptococcosis in marine mammals.

Case	Mammal Species	Diagnosis	Species/Variety	Outcome (Treatment, Necropsy)	Location	Date	References
1	Whale (*Eubalaena australis*) neonate	Skin lesion culture positive for *Candida zeylanoides*	*Cryptococcus neoformans*	Necropsy	South Africa	2009	[81]
2	Spinner dolphin (*Stenella longirostris*)	Skin, lymph nodes and several organs (stomach, adrenal gland, kidney and spleen)	*Cryptococcus gattii*—VGI	Necropsy	Hawaii	2010	[79]
3	Striped Dolphin (*Stenella coeruleoalba*)	Pulmonary cryptococcosis (lung and mediastinal lymph gland) and gastric mucosa. Stomach with moderate number of nematodes (Anisakis simplex)	*Cryptococcus* spp.	Necropsy	Western Australia	1985	[80]
4	Bottlenose dolphin (*Tursiops truncatus*)	Pulmonary Cryptococcosis Bronchopneumonia with pleuritis	*Cryptococcus gattii*	Necropsy Itraconazole therapy	California, USA	2000	[86]
5	Atlantic bottlenosed dolphin (*Tursiops truncatus*)	Pulmonary cryptococcosis	*Cryptococcus* spp.	Necropsy	Not reported	1978	[83]
6	Dall’s porpoise (*Phocoenoides dalli*)	Pulmonary cryptococcosis with generalized lymphadenopathy	*Cryptococcus gattii*	Necropsy	Vancouver Island, Canada	2000–2001	[52]
7	Dall’s porpoise (*Phocoenoides dalli*)	Pulmonary cryptococcosis with generalized lymphadenopathy	*Cryptococcus gattii*	Necropsy	Vancouver Island, Canada	2000–2001	[52]
8	Dall’s porpoise (*Phocoenoides dalli*)	Pulmonary cryptococcosis with generalized lymphadenopathy	*Cryptococcus neoformans*	Necropsy	Vancouver Island, Canada	2000–2001	[52]
9	Dall’s porpoise (*Phocoenoides dalli*)	Pulmonary cryptococcosis with generalized lymphadenopathy	*Cryptococcus neoformans*	Necropsy	Vancouver Island, Canada	2000–2001	[52]
10	Dall’s porpoise (*Phocoenoides dalli*)	Pulmonary cryptococcosis with generalized lymphadenopathy	*Cryptococcus neoformans*	Necropsy	Vancouver Island, Canada	2000–2001	[52]
11	Harbor porpoise (*Phocoena phocoena*)	Pulmonary cryptococcosis with generalized lymphadenopathy	*Cryptococcus neoformans*	Necropsy	Vancouver Island, Canada	2000–2001	[52]
12	Adult and foetus Harbor porpoise (*Phocoena phocoena*)	Maternal–Foetal transmission of *Cryptococcus*	*Cryptococcus gattii*	Necropsy	USA	2007	[88]
13	Porpoise	Not reported	*Cryptococcus gattii*—VGI	Culture. Multi-Locus Sequence Typing (MLST)	Pacific Northwest Coast, Canada	2010	[85]
14	Porpoise	Not reported	*Cryptococcus gattii*—VGIIa	Culture. Multi-Locus Sequence Typing (MLST)	Pacific Northwest Coast, Canada	2010	[85]
15	Porpoise	Not reported	*Cryptococcus gattii*—VGIIa	Culture. Multi-Locus Sequence Typing (MLST)	Pacific Northwest Coast, Canada	2010	[85]
16	Porpoise	Not reported	*Cryptococcus gattii*—VGIIa	Culture. Multi-Locus Sequence Typing (MLST)	Pacific Northwest Coast, Canada	2010	[85]
17	Dall’s porpoise (*Phocoenoides dalli*)	Pyogranulomatous pneumonia and lymphadenitis	*Cryptococcus gattii*	Necropsy	California, USA	2014	[89]
18	California sea lion (*Zalophus californianus*)	Systemic mycosis (skin lesions, dermal nodules, severe lymphadenopathy)	*Cryptococcus* spp.	Necropsy—combination of antibiotic (enrofloxacin, amoxicillin trihydrate-clavulanate potassium), low-dose steroid (prednisone, furosemide), anti-inflammatory (carprofren), oral diphenhydramine, itraconazole and voriconazole. Growth of *Escherichia coli*.	USA	2012	[84]
19	Juvenile Californian sea lion (*Zalophus californianus*)	Pneumonia with concurrent fungal and bacterial infection	*Cryptococcus albidus*	Necropsy	California, USA	2009	[82]
20	Young (3 weeks) harbor seal (*Phoca vitulina*)	Systemic cryptococcosis: generalized lymphadenopathy, bronchopneumonia, meningoencephalitis, fungemia	*Cryptococcus gattii*—VGIIa	Necropsy	Vancouver Island, Canada	2016	[87]
21	Dolphin	Not reported	*Cryptococcus gattii*—VGI	Culture. Multi-Locus Sequence Typing (MLST)	Pacific Northwest Coast, Canada	2010	[85]
22	Dolphin	Not reported	*Cryptococcus gattii*—VGI	Culture. Multi-Locus Sequence Typing (MLST)	Pacific Northwest Coast, Canada	2010	[85]
23	Dolphin	Not reported	*Cryptococcus gattii*—VGI	Culture. Multi-Locus Sequence Typing (MLST)	Pacific Northwest Coast, Canada	2010	[85]
24	22× living harbor seals (*Phoca vitulina)*, juvenile	Not reported	Nasal swabs (19×) and lung culture (3×) negative for *Cryptococcus* spp.	Culture	Vancouver Island, Canada	Between February and August 2004	[50]

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
