# Peer review of "Cryptococcus in Wildlife and Free-Living Mammals"

_jof, 2021, doi:10.3390/jof7010029_

Round 1

Reviewer 1 Report

This paper reviews Cryptococcus infections in a range of free-living wildlife animal species with the hypothesis that some of these animal species may act as disease sentinels for human infections. Overall, I think paper reads well and adequately covers the literature. However, I have some comments.

  • I like the different sections of the paper, especially the pictures. Do you have any pictures of marine mammals with Cryptococcus? Is there enough information to also add a treatment section? Do most of these animals succumb to the infection or are any of them treated? If so, how successful is the treatment?
  • What does “WA” refer to? This abbreviation first appears on line 313 and is not defined.
  • Figure 4 is hard to read. The Table appears to be pixelated, making the words fuzzy and hard to read.

Minor points:

  • On line 191, remove “but” from the sentence. It is not needed.
  • One lines 268-269, this sentence is confusing. I suggest restructuring it as follows: “….may predispose captive cheetah kept under stressful conditions to cryptococcal infections.”
  • On line 361, “commonly” should be “common”.
  • On line 377, “tree-ranging, miale” should be “free-ranging, male”.
  • On line 394, add “the” before “Mediterranean”.
  • On line 451, remove the parenthesis before “Instead”.
  • On line 459, “manifest” should be “manifested”.

Author Response

Reviewer1

Your comment:

This paper reviews Cryptococcus infections in a range of free-living wildlife animal species with the hypothesis that some of these animal species may act as disease sentinels for human infections. Overall, I think paper reads well and adequately covers the literature. However, I have some comments. I like the different sections of the paper, especially the pictures

Thank you for your kind comments.

Your comment:

Do you have any pictures of marine mammals with Cryptococcus? Is there enough information to also add a treatment section? Do most of these animals succumb to the infection or are any of them treated? If so, how successful is the treatment

We are sorry but we do not have pictures about marine mammal cases other than the ones published elsewhere. Such cases are rare in Australia and Italy – and we have not been fortunate enough to have any direct or indirect involvement in the diagnosis and management of cases – hence we have no photos of our own.

There is very little information on treatment of marine mammals in the literature. The data available from references we consulted are summarized in Table 2 (Cryptococcosis in marine mammals). This is equally the case for treatment of the other species. The same drugs are used in all species – fluconazole and amphotericin B – but the practicalities of giving large doses to different species is quite involved and beyond the scope of this manuscript. For example, in horses – there is a choice of IV amphotericin daily versus fluconazole orally, although ideally both could be given at the same time. When treating marine mammals, difficult in securing IV access on an ongoing basis usually means therapy consists of azoles which can be hidden in food. The technical issues involved would require many paragraphs to cover – although such detail exists, especially in relation to the koala, where the treatment of individual animal in captivity is commonly done and justified.

According to the literature, the picture of mycosis in marine mammals raised in the last two decades, with an increasing of mucormycotic infection incidence (like what has been seen in human). Aspergillosis is the other very important systemic mycoses in sea mammals. Thus, most of treatments described are about fungal infection caused by fungi in general rather than not Cryptococcus. Overall, antifungal drugs seem having only modest success in reducing the high mortality rates associated with invasive mycoses in marine mammals and caused in large part by delays in disease diagnosis and fungal identification, combined with the difficulty in giving prolonged intravenous therapy using amphotericin B. The new formulations of experimental amphotericin B that can be given orally would be a huge advantage in management of marine mammals with cryptococcosis, but there is currently no availability.

Your comment

What does “WA” refer to? This abbreviation first appears on line 313 and is not defined.

Reviewer 1 is right. WA stands for Western Australia. This abbreviation is now provided in the updated MS. SORRY!

Your comment

Figure 4 is hard to read. The Table appears to be pixelated, making the words fuzzy and hard to read.

Thank you for the comment. Now Figure 4 is fixed and re-named Figure 5 in the updated MS.

Minor points:

The text was revised and re-written in some parts, where sentences were found using computer algorithms to be too much like ones already published elsewhere by the editor. The follow points were amended, as suggested, during the rewriting process. THANK YOU.

On line 191, remove “but” from the sentence. It is not needed.

One lines 268-269, this sentence is confusing. I suggest restructuring it as follows: “…. may predispose captive cheetah kept under stressful conditions to cryptococcal infections.”

On line 361, “commonly” should be “common”.

On line 377, “tree-ranging, male” should be “free-ranging, male”.

On line 394, add “the” before “Mediterranean”.

On line 451, remove the parenthesis before “Instead”.

On line 459, “manifest” should be “manifested”.

Reviewer 2 Report

This is an interesting review about Cryptococcus in wildlife and free-living mammals. I only have a few minor comments.

  1. It would be useful to include description of Cryptococcus cells in the tissues of these animals (histological characterization) and make a brief comparison with those found in humans. For instance, Cryptococcus of atypical morphology is more likely to be found in the upper-respiratory track in animals. Is there any difference in capsule, cell size, cell shape of Cryptococcus cells found in different animals comparing to humans?
  2. A few references could be included in the discussion on life cycle on cryptococcal pathogenesis and the impact of different morphotype on host immune responses. For example, Zhao et al. ARM review on cryptococcal life cycle (2019), Zhai et al research on host immune responses elicited by yeast and hyphae (2015, mBio), Christina Hull's recent research on the difference of immune responses to yeast and spores.
  3. There are multiple places where the sentences miss periods. A few spelling errors (that/than etc).
  4. Glass glare in Figure 2C is hard to detect.
  5. It would be nice to include histological images in addition to the animals and whole organs.

Author Response

Reviewer 2

Your comment

This is an interesting review about Cryptococcus in wildlife and free-living mammals. I only have a few minor comments.

Thank you for your kind comments and words

Your comment

It would be useful to include description of Cryptococcus cells in the tissues of these animals (histological characterization) and make a brief comparison with those found in humans. For instance, Cryptococcus of atypical morphology is more likely to be found in the upper-respiratory track in animals. Is there any difference in capsule, cell size, cell shape of Cryptococcus cells found in different animals comparing to humans?

We agree that this is a very interesting point. Your example is in relation to the respiratory tract. We have always been of the view that the morphology of Cryptococcus in skin is very different to in other tissues. We think this is more to do with the physiological features of location – things like temperature, oxygen tension etc – and probably this accounts for the differences the reviewer is alluding to, rather than due to species per se. But to do justice to this discussion point – we would need to systematically review the literature and probably there would not be sufficient space in THIS paper to do this.

Your comment

A few references could be included in the discussion on life cycle on cryptococcal pathogenesis and the impact of different morphotype on host immune responses. For example, Zhao et al. ARM review on cryptococcal life cycle (2019), Zhai et al research on host immune responses elicited by yeast and hyphae (2015, mBio), Christina Hull's recent research on the difference of immune responses to yeast and spores.

We thank for kind suggestions. We have added Zhao et al., 2019 reference, but a detailed discussion of concepts like titan cells and tubular mitochondria is really beyond the scope of this general overview. It’s a worthy topic for discussion, but we just have run out of words for the word limit of this special issue.

Your comment

There are multiple places where the sentences miss periods. A few spelling errors (that/than etc).

The text was revised and re-written in some parts and amended as suggested

Your comment

Glass glare in Figure 2C is hard to detect.

Probably you are correct – but to a person use to looking at cheetahs – they tend to look at you and follow you with their eyes. Whereas at presentation – the cheetah would just stare into the distance. The photograph sort of captures this phenomenon.

Your comment

It would be nice to include histological images in addition to the animals and whole organs.

Thank you for kind suggestions. Histological photos about Cryptococcosis in koalas (Figure 2) were included in the updated MS. 
